

# The effect of carbonic anhydrase on foraminiferal Mg/Ca

Siham De Goeyse[1], Chiara Lesuis[1,2], Gert-Jan Reichart[1,3] and
Lennart de Nooijer[1]

[1] Ocean Sciences, NIOZ—Netherlands Institute for Sea Research, Texel, Netherlands
[2] Institute for Sanitary Engineering, Water Quality and Solid Waste Management, University of
Stuttgart, Stuttgart, Germany
[3] Geosciences, Utrecht University, Utrecht, Netherlands

## ABSTRACT

Marine biogenic calcium carbonate production plays a role in the exchange of $CO_2$ between ocean and atmosphere. The effect of increased $CO_2$ on calcification and on the resulting chemistry of shells and skeletons, however, is only partly understood. Foraminifera are among the main marine $CaCO_3$ producers and the controls on element partitioning and isotope fractionation is the subject of many recent investigations. The enzyme carbonic anhydrase (CA) was, for example, shown to be vital for $CaCO_3$ deposition in benthic foraminifera and indicates their ability to manipulate their intracellular inorganic carbon chemistry. Here, we tested whether CA affects the partitioning of Na, Mg and Sr in the perforate, large benthic, symbiont-bearing foraminifer *Amphistegina lessonii* by addition of the inhibitor acetazolamide (AZ). The effect of dissolved $CO_2$ on the effect of CA on element partitioning was also determined using a culturing setup with controlled atmospheric carbon dioxide levels (400–1,600 ppm). Results show that inhibition by AZ reduces calcification greatly and that $CO_2$ has a small, but positive effect on the amount of calcite formed during the incubations. Furthermore, the inhibition of CA activity has a positive effect on element partitioning, most notably Mg. This may be explained by a (n indirect) coupling of inorganic carbon uptake and inward calcium ion pumping.

## INTRODUCTION

Calcifying foraminifera are marine protists responsible for approximately half of the open ocean calcium carbonate production (*Schiebel, 2002*). Their fossil shells are also widely used as tools for reconstructing the paleoenvironment and paleoclimate. Incorporation of minor elements in their shell is used to *e.g.*, reconstruct temperature and to trace back the composition of the seawater in which they calcified. One of the most commonly used proxies in paleoceanography, next to stable oxygen isotopes, is the Mg/Ca ratio of foraminiferal shells. The amount of Mg incorporated in the calcitic shell has been shown to primarily reflect sea water temperature (*Nürnberg, Bijma & Hemleben, 1996*; *Anand, Elderfield & Conte, 2003*), but is also influenced by so-called vital effects (*Erez, 2003*; *Bentov & Erez, 2006*; *De Nooijer et al., 2014*). Although Mg/Ca is one of the most robust temperature proxies known, foraminiferal life processes such as their metabolism and the

Corresponding author
Lennart de Nooijer,
ldenooijer@nioz.nl

biology involved in biomineralization is offsetting the thermodynamic relation between Mg/Ca partitioning and temperature. Such effects are species-specific and are apparent from the variability in average Mg/Ca (*e.g.*, *Bentov & Erez, 2006*; *Sadekov et al., 2016*; *Van Dijk, De Nooijer & Reichart, 2017*), variations in Mg/Ca-temperature calibrations (*Toyofuku et al., 2011*; *Wit et al., 2012*), and offsets both in ratios and Mg/Ca-temperature sensitivities between foraminifera and inorganically precipitated calcites (*Bentov & Erez, 2006*).

Understanding foraminiferal calcification is also important because of its important role in the carbon cycle because of the continuous production of calcitic shells by planktonic species in the upper water column and subsequent downward transport. It has been estimated that up to 50% of the open ocean biogenic calcium carbon flux consists of foraminiferal shells (*Schiebel, 2002*), and a collective response in calcification to climate change may thus have an impact on sea water carbonate chemistry. Calcium carbonate precipitation in the ocean shifts the carbonate chemistry in such a way that enhanced calcium carbonate precipitation increases surface water $p$CO$_2$ (*Zeebe & Wolf-Gladrow, 2001*), whereas water column dissolution reduces $p$CO$_2$ at depth, thereby *de facto* counter acting the biological carbon pump. This balance between organic and inorganic carbon transport, which is called the rain ratio, thereby plays a major role in setting atmospheric $p$CO$_2$ under natural conditions. This implies that reduced and/or enhanced foraminiferal biomineralization as a consequence of ongoing anthropogenic CO$_2$ emissions and associated ocean acidification potentially could provide an important negative or positive feedback (*e.g.*, *Sarmiento et al., 2002*; *Feely et al., 2004*). Since already 25% of the anthropogenic CO$_2$ has been absorbed by the oceans since the industrial revolution (*e.g.*, *Sabine & Tanhua, 2010*), understanding such feedback is clearly urgent and of great importance.

Foraminiferal calcification has been studied in detail to understand so-called vital effects and also to understand the impact of changes in the sea water carbonate system on calcification. Several studies looked into the way foraminifera modify the chemistry at the site of calcification (SOC) and also outside the SOC. Foraminifera are known to strongly affect the pH, both inside (*Bentov, Brownlee & Erez, 2009*; *De Nooijer et al., 2009*) and outside the SOC (*Rink et al., 1998*; *Köhler-Rink & Kühl, 2000*, *2005*; *Glas, Langer & Keul, 2012*; *Toyofuku et al., 2017*). It has been argued that active proton pumping, visible from the pH gradient between the SOC and outside the foraminifer, is directly coupled to inward Ca$^{2+}$ transport (*Toyofuku et al., 2017*). This would involve exchange of two protons being pumped out of the SOC in exchange for one Ca$^{2+}$ ion entering the SOC (*Nehrke et al., 2013*; *Toyofuku et al., 2017*).

The H$^+$-pumping has also been suggested to cause an inward diffusion of CO$_2$ as a result of the steep pH gradient (~9.0 at the site of calcification, ~6.5 outside low-Mg/Ca species; *Toyofuku et al., 2017*). The high pH in the calcifying fluid may also attract CO$_2$ from the foraminifer's cytosol, which has been invoked to explain low calcite $\delta^{13}$C (*Schmiedl et al., 2004*). Counterintuitively, this implies that a higher ambient $p$CO$_2$ may stimulate calcification by foraminifera as it would enhance CO$_2$ fluxes, explaining the positive impact of CO$_2$ on calcification observed in some culturing experiments

(*Haynert et al., 2014*). Other studies, however, indicated that growth was hampered by elevated $CO_2$ levels (*Hikami et al., 2011*). This apparent contradiction between studies may be explained by the involvement of other inorganic carbon species. Bicarbonate ions ($HCO_3^-$) may be taken up and converted by the enzyme carbonic anhydrase (CA), an enzyme catalyzing the reversible hydration of $CO_2$ to form $H_2CO_3$ with the direction of the reaction entirely dictated by pH (*Lindskog, 1997*). Indirectly, this may also help to convert the $CO_2$ to $CO_3^{2-}$ after diffusing to the site of calcification in foraminifera (*De Goeyse et al., 2021*). Although the pH inside the SOC has been measured to be close to 9 (*De Nooijer et al., 2009*), conversion of $CO_2$ without CA would be the rate limiting step in calcification (see *e.g.*, *Chen, Gagnon & Adkins, 2018* for the role of CA in coral calcification). This has been highlighted in other marine calcifying organisms where carbonic anhydrase was found to be an integral part of the biomineralization pathway (*Bertucci et al., 2013*; *Medaković, 2000*; *Müller et al., 2013*; *Le Roy et al., 2014*; *Wang et al., 2017*).

Here, we investigate element pumping by foraminifera during biomineralization using acetazolamide to block the functioning of carbonic anhydrase and through varying carbonate chemistry in controlled growth experiments. We focused on the foraminiferal species *Amphistegina lessonii*, a large benthic, symbiont-bearing foraminifer that forms a calcite shell as this species has been investigated in detail for its biomineralization pathways and impact of different environmental factors (*Raja et al., 2005*; *Raja, Saraswati & Iwao, 2007*; *Segev & Erez, 2006*; *De Nooijer et al., 2017*; *Geerken et al., 2018*; *Levi, Müller & Erez, 2019*; *Dämmer et al., 2021*). We measured the minor element composition of the shells produced under contrasting conditions to elucidate calcification mechanisms, elemental pathways and carbon uptake.

## MATERIALS AND METHODS

### Foraminifera collection and incubation

Sediments were collected from the Indo-Pacific coral reef aquarium at Burgers' Zoo in September 2021. Adult specimens of *A. lessonii* were isolated from these sediments and were incubated in Petri dishes with filtered seawater (Whatmann, 0.3 μm pore filters) for 1 week at 18 °C with a 12 h light/dark cycle and no added food. After this week they were fed living *Dunaliella salina* and transferred to a 26 °C incubator with a 16/8 h light/dark cycle to trigger reproduction. Cultures were monitored daily and when reproduction occurred, juveniles were stored at 18 °C and transferred to experimental conditions within 1–3 days. With a rapid addition of chambers in these first days after reproduction, it is unavoidable that the specimens that were incubated already added a few chambers; inspection revealed that these were never more than five chambers.

Experimental incubation took place at a constant temperature of 24 °C. All juveniles were exposed to a 12 h light/dark cycle (~180 μmol photons m$^{-2}$ s$^{-1}$). Juveniles were incubated in 200 mL tissue culture flasks in one out of four incubators with its own, constant $p$CO$_2$ set to a target value of 400, 800, 1,200 or 1,600 ppm (Fig. 1). In these incubators, $p$CO$_2$ was kept as close to the set target value as possible. Inflowing air was scrubbed using soda lime to remove any $CO_2$ and mixed with pure $CO_2$ to achieve the target experimental $p$CO$_2$ value. Inside the incubator, $p$CO$_2$ was constantly monitored and

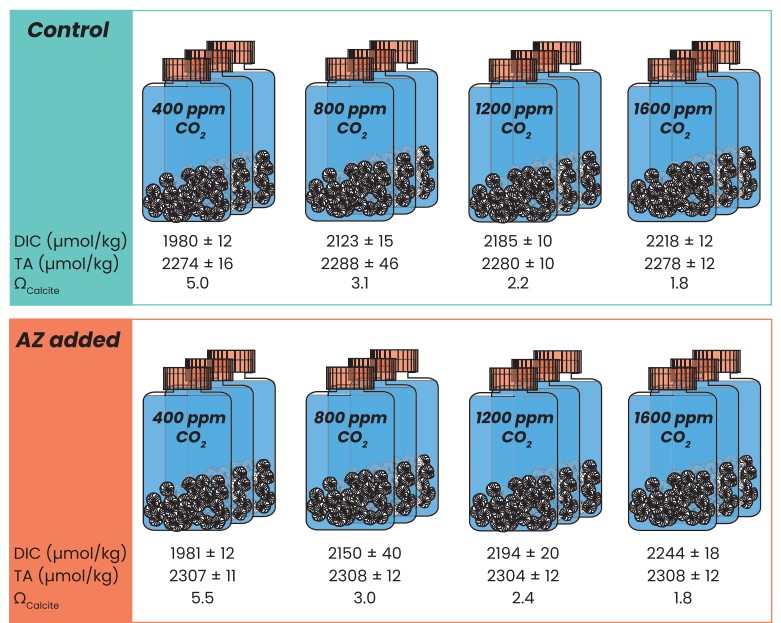

**Figure 1 Experimental setup.** Schematic overview of all incubations with $p\mathrm{CO_2}$ ranging from 400 to 1,600 ppm, in control (blue) conditions and with 8 μM AZ (in orange), carried out in triplicate per condition. Every flask contained 30–40 specimens of *Amphistegina lessonii*. Measured DIC and TA are indicated below each triplicate, including the calculated saturation state with respect to calcite.

the balance of the incoming $\mathrm{CO_2}$-free air: pure $\mathrm{CO_2}$ was automatically adjusted (*Dämmer et al., 2023*). Throughout the experiment, measured $p\mathrm{CO_2}$ varied within ±30 ppm (maximum deviation) of the target value. To prevent evaporation of the culture media, inflowing air was saturated with water by bubbling it through MilliQ before entering the incubator.

Before each experiment, culture flasks containing 200 mL North Atlantic seawater (NASW with a salinity of 36) were allowed to equilibrate at set experimental $p\mathrm{CO_2}$ conditions for between 24 h and several weeks. Following reproduction, 30–40 juvenile specimens were placed into each of these pre-equilibrated flasks. For the control conditions, the flasks contained only sea water and the juvenile specimens. For the experimental conditions, 18 μL of 90 mM acetazolamide (AZ) dissolved in DiMethyl SulfOxide (DMSO) was added to each flask at the start of incubation, resulting in a final AZ concentration of 8 μM. For each condition, incubation was carried out in triplicate, resulting in the incubation of 90–120 specimens per condition (Fig. 1).

The culture media were replaced with fresh, pre-equilibrated sea water twice during the incubation period: after 2 and 4 weeks. After 2 weeks, the incubated juveniles were also fed living *D. salina*. Before feeding, the *D. salina* culture was washed three times with NASW, each time centrifuging for 3 min at 3,000 RPM. *A. lessonii* were fed 200 μL *D. salina* suspension. After 4 weeks, the water was changed once more, and no food was added since the bottoms of all flasks still visibly contained algae. After 5 weeks, the incubation was terminated by submerging the foraminifera in MilliQ. Thereafter, specimens were left to dry and stored at 4 °C until further analysis.

## Alkalinity, DIC and nutrient analysis

Dissolved inorganic carbon (DIC), total alkalinity (TA) and nutrient samples of the incubation medium were collected after 2, 4 and 5 weeks of incubation. For DIC measurements, 12 mL of culture media were taken, poisoned with 15 μL $HgCl_2$ and stored in airtight vials at 4 °C until analysis. DIC measurements were carried out on a QuAAtro Continuous Segmented Flow Analyzer (*Stoll et al., 2001*), with an analytical error of ~2.5 μM. For TA, 5 mL subsamples poisoned with $HgCl_2$, were analysed on a QuAAtro Continuous Segmented Flow Analyzer (*Sarazin, Michard & Prevot, 1999*), with typical analytical error of <0.5% of the average. For nutrient analysis, 5 mL samples were taken and stored at −18 °C. Nutrient concentrations for $PO_4$, $NH_4$, $NO_2$ and $NO_3$ were analyzed on a TrAAcs 800 autoanalyzer (*Murphy & Riley, 1962*; *Helder & De Vries, 1979*; *Grasshoff, Kremling & Ehrhadt, 1999*). For these analyses, a nutrient cocktail was used to monitor precision, showing reproducibility better than 1.5%, but typically 0.7% from the average. Other carbonate system parameters (pH, $pCO_2$, and carbonate speciation) were calculated using PyCO2SYS (*Humphreys et al., 2022*).

## LA-ICP-MS preparation and measurement

After incubation, all specimens underwent an oxidative cleaning treatment to remove organic matter. For each replicate, foraminifera were transferred to a 0.5 mL Eppendorf tube, resulting in a total amount of 30–40 specimens per tube. To each tube, 250 μL freshly prepared alkali buffered 0.33% $H_2O_2$ solution in 0.5M $NH_4OH$ (0.5 ml and 49.5 ml, respectively) was added. All tubes were then heated in a water bath at 95 °C for 10 min, and placed in an ultrasonic bath (80 kHz, 50% power) for 30 s. These steps were repeated twice. Afterwards, all samples were rinsed three times with MilliQ and dried before analysis.

Elemental ratios in the calcite formed during the incubation period (5–15 chambers) were measured using Laser Ablation Inductively Coupled Plasma Mass Spectrometry (LA-ICP-MS). To ensure that the measured chambers were formed during the incubation period, the laser was focused on each of the final three chambers of specimens cultured under control conditions, and on the final chamber of specimens cultured with AZ.

For the control incubations, 10 specimens for each replicate per $pCO_2$ were analyzed, resulting in a total of 120 measured specimens. For the specimens incubated with AZ, 10 specimens of only one replicate per $pCO_2$ were measured (40 specimens total, 10 per replicate), due to the reduced growth in this treatment. Cleaned specimens were placed on a stub with double-sided tape. Chambers were ablated with in a NWR193UC TV2 dual-volume chamber using a circular laser (193 nm) spot with a diameter of 60 μm and a pulse repetition of 6 Hz and an energy density of $1.00 \pm 0.05$ $Jcm^{-2}$. The aerosol produced during the ablation was transported to a quadrupole ICP-MS (Thermo Fisher Scientific iCAP-Q; Thermo Fisher Scientific, Waltham, MA, USA) on a helium flow with a flow rate of 0.6 L $min^{-1}$, with 0.4 L $min^{-1}$ argon make-up gas being added before entering the ICP torch. Elemental quantification was based on cosmic relative abundances of the isotopes considered and using NIST SRMs 612 and 610 (*Jochum et al., 2011*) and using matrix matched standards MACS3, JcT and JcP and NFHS2-NP. At the start of each series and

after every 10–15 samples, the NIST SRM 610 was analyzed. Intensities from the N610 were used to convert the mass intensity ratios (w/t Ca) of the samples to mol/mol.

NFHS-2-NP standard was used to monitor drift after every fourth sample. NFHS-2-NP is the preferred homogeneous reference material for drift monitoring at NIOZ as it shows lower gas blanks for some elements (*e.g.*, Na) than NIST SRM 610 (*Boer et al., 2022*). The calibrated data of NFHS-2-NP using NIST SRM 610 without drift correction were used to calculate drift correction factors (*i.e.*, cps *vs* time) over the course of one run. The first four NFHS-2-NP standards directly measured after the calibration standards were assumed to undergo no drift (*i.e.*, their average intensity was 100%). Although instrumental drift was small, drift corrections were made to improve the accuracy.

The following mass ratios were measured for element quantification: $^{23}Na/^{43}Ca$, $^{25}Mg/^{43}Ca$, $^{88}Sr/^{43}Ca$. Repeatability, *i.e.*, variation in the NFHS2-NP was 2.4% (RSD) for Na/Ca, 0.9% for Mg/Ca, and 0.8% for Sr/Ca. Accuracy was between 99% and 101% for Na/Ca, Mg/Ca and Sr/Ca based on repeated measurements of the MACS3 standard. Throughout the ablation photographs of the ablation spot were taken at 2 s intervals. Ablation profiles were analyzed in MATLAB and examined next to these photographs. The interval selected for elemental quantification was automatically assigned based on the period during which an increase and decrease in Ca was observed (begin and end shell carbonate during ablation). These intervals were later manually adjusted when needed to ensure that the measured interval corresponded to the time period during which ablation photographs showed a hole appearing in the foraminiferal shell wall. Ablation profiles were excluded when the photographs showed no hole appearing during the ablation period, or when no convincing window of Ca increase and subsequent decrease could be detected.

## Size

To estimate the size of the cultured specimens, cleaned specimens were placed on a scaled microscope slide and photographed using a stereomicroscope. The cross-sectional area of these specimens was determined using dedicated software (Fiji, DOI: 10.1038/nmeth. 2019). Pictures were converted to eight-bit and the pixel value cut-off threshold was adjusted manually to ensure that the chosen threshold corresponded to the specimen's outline. In cases where the lighting made this difficult, specimens' outlines were traced manually. All pictures were then converted to binary images and converted to mask. Using the scale on the pictures, size analysis was carried out using the 'Analyze Particles' module within the Fiji software (size 0.05-Infinity, circularity 0.05–1.00) for the control specimens, and 'Analyze Particles' (size 0.005–Infinity, circularity 0.05–1.00) for specimens incubated with AZ.

## Statistical analysis

To assess the (dis)similarity between replicates, an analysis of the difference between the means was carried out for all replicates of each condition. Within each experimental condition, a Shapiro-Wilk analysis was carried out on each replicate to check if data was distributed normally. A Levene test was conducted to assess equality of variance between

replicates. For TA, two pairs of replicates (1,200 ppm $CO_2$/AZ added; 400 ppm $CO_2$/control) had slightly different average values (t-test assuming unequal variances, $p < 0.05$); for the DIC analysis, one pair within a treatment had significantly different average values (1,600 ppm $CO_2$/AZ added). For the nutrient concentrations there were no significant differences in the average concentrations between the replicates within any of the eight treatments (t-test assuming unequal variances, $p > 0.05$ in all cases).

To test for the effect of $pCO_2$ on El/Ca, we performed an ordinary least sum of squares (OLS) regression analysis assuming a linear response model. This was done in python using the 'statsmodels' library with $CO_2$ as the independent and El/Ca as the dependent variable.

As the majority of the data did not follow a normal distribution, non-parametric methods were used for further data analysis. This was done for both end size analysis and LA-ICP-MS results. For LA-ICP-MS, outliers were identified using the interquartile range (IQR). If data points fell outside of $\pm 1.5 * $ IQR, and visual inspection of the LA-ICP-MS depth profile indicated an error in the measurements, the data points were excluded from further analysis. This resulted in the exclusion of 12 out of 120 data points for the control group and three out of 38 data points for the AZ group.

To relate the obtained Mg/Ca to Na/Ca, a regression analysis was performed. This analysis was done using an ordinary least sum of squares (OLS) assuming a power function (Na/Ca = a $* e^{Mg/Ca}$ + b). We use the package for Python "scipy" with the module "curvefit" (*Virtanen et al., 2020*). We report the $p$-values of the correlations and the equations, based on correlating Mg/Ca and Na/Ca of all conditions. We calculated the mean and standard errors (SE) for the regression parameters and the shown confidence intervals were calculated from these standard errors.

## Comparison between solution- and LA-ICP-MS data

To test the effect of laser ablation (LA) on the obtained Na/Ca, Mg/Ca and Sr/Ca, we measured a number of specimens after dissolution using a sector field-ICP-MS (Element2; Thermo Scientific, Waltham, MA, USA). To do this a volume of 30 μL from every sample was pre-scanned for its calcium concentration ([$Ca^{2+}$]). Based on this pre-scan, all samples' element concentrations were determined from matched [$Ca^{2+}$]. For boron, magnesium and strontium, masses ($^{11}$B, $^{25}$Mg and $^{88}$Sr, respectively) were measured in low resolution, whereas sulfur (mass 32) was analyzed in medium resolution to avert interference by $^{16}$O$^{16}$O. Iron ($^{56}$Fe) was measured at high resolution to avoid interference of $^{16}$O$^{40}$Ar. In all samples, $^{43}$Ca was monitored to calculate the elemental ratios. This was done using known cosmic relative abundances. Measured samples alternated with 0.1 MHNO$_3$ during the runs for effective wash-out and to avoid carry-over of between samples. All samples were compared to five ratio calibration standards (*De Villiers, Greaves & Elderfield, 2002*). To monitor drift throughout measurements, we included together with three additional standards: NFHS-2-NP (*Boer et al., 2022*), JCt (*Tridacna gigas* giant clam; S/Ca = 0.57 ± 0.04 mmol/mol, B/Ca = 0.23 ± 0.01 mmol/mol, Mg/Ca = 1.21 ± 0.01 mmol/mol, $n = 7$, ± indicates 1 $1\sigma$ SD) and JCp (*Porites* sp. coral; S/Ca = 5.79 ± 0.14 mmol/mol, B/Ca = 0.50 ± 0.01 mmol/mol, Mg/Ca = 4.05 ±

0.04 mmol/mol, $n = 7$, ± indicates 1 $1\sigma$ SD). Since the number of replicates was small for the solution-ICP-MS data and since there were no trends of El/Ca with $p$CO$_2$, the El/Ca for the four conditions (both control and when AZ was added) were pooled. A t-test assuming equal variances and $\alpha = 0.05$ showed no significant differences in the averages of Na/Ca and Sr/Ca between the two types of analysis. For Mg/Ca, solution-ICP-MS ratios were slightly lower than the LA-ICP-MS obtained ratios (Tables S2 and S3).

## Rayleigh distillation model

To test if the observed trend in Na/Ca and Mg/Ca may be explained by Rayleigh distillation, a simple model was constructed. Ions of magnesium, calcium and sodium are assumed to be incorporated from a fluid with a finite volume into a calcite crystal. The partition coefficient for Mg and Na are described by:

$$K_{Mg} = \left([Mg^{2+}]/[Ca^{2+}]_{calcite}\right)/\left([Mg^{2+}]/[Ca^{2+}]_{fluid}\right) \tag{1}$$

where $K_{Mg}$ is the partition coefficient for magnesium, which is proportional to the ratio of the concentrations of Mg$^{2+}$ and Ca$^{2+}$ in the calcite, divided by that ratio in the fluid from which the CaCO$_3$ precipitates.

If incorporation removes Mg$^{2+}$ and Ca$^{2+}$ in a fixed ratio from a finite volume, the ratio of those two ions remaining in the fluid changes. This is the simplest form of Rayleigh distillation and is described by Eq. (2):

$$D_{Mg} = (1 - f^{KMg})/(1 - f) \tag{2}$$

where $D_{Mg}$ is the distribution coefficient for magnesium and f the fraction of the Ca remaining in the reservoir. If f is close to 1, almost no ions are precipitating from the fluid, while f = 0 indicates that all calcium ions are removed from the calcifying reservoir. In case of the latter, distribution coefficients are close to 1; with a large fraction remaining, distribution coefficients are close the partition coefficients ($D_{Mg} \approx K_{Mg}$). The partition coefficient (K) is here approximated by the partitioning of an ion during inorganic CaCO$_3$ precipitation and is determines part of the overall seawater-foraminiferal calcite partitioning (D). The difference between D and K can be seen as the cellular controls that create a calcifying fluid in which the ions' concentrations are different than those in seawater.

For Na, we assume that Eqs. (1) and (2) are similar. With a system with both Mg and Na being incorporated into the calcite, the ratio of the two ions in the calcite is a function of the ratio of $K_{Mg}$ and $K_{Na}$. Since these coefficients are also affected by precipitation rate, presence of other ions/organic compounds, CaCO$_3$ phase transformations, *etc.*, we used a range of inorganic partition coefficients to calculate the ratio of foraminiferal Mg/Ca *vs* Na/Ca, which essentially reflects the ratio between the inorganic partition coefficients. This is because the factor f (Eq. (2)) has to be the same for both Mg- and Na-incorporation. Since the (initial) composition of the calcifying fluid is essentially unknown, we assume a ten-fold higher (Ca$^{2+}$) then seawater at the site of calcification. This remains a rough estimate, but an enrichment relative to seawater is likely given the 2H$^+$/Ca$^{2+}$ exchange during calcification for rotaliid foraminifera (*Toyofuku et al., 2017*).

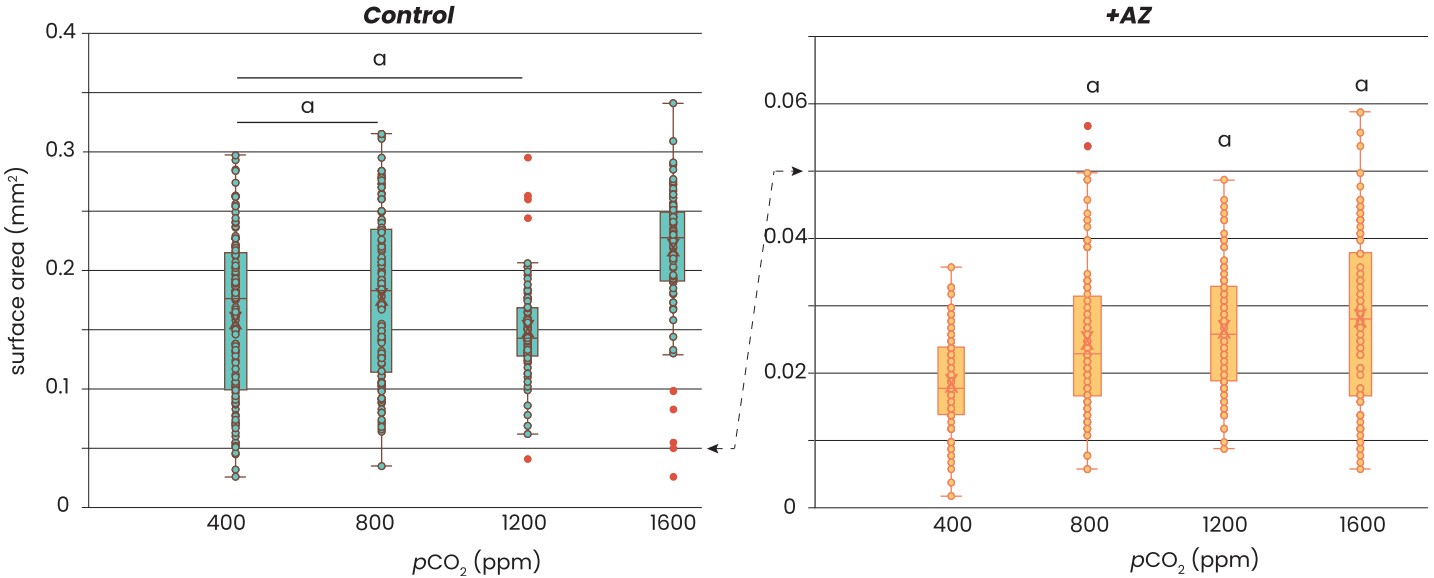

**Figure 2 Relation between foraminiferal growth and CO₂.** Recorded size of the foraminifera cultured at different $pCO_2$'s is plotted without (left) and with addition of AZ (right): note the difference in the scale of the y-axis. Dots indicate individual foraminifera: outliers are indicated in red and are lower than 1.5 times the lower quartile or higher than 1.5 times the third quartile. Box-whisker plots indicate the average ('X'), the median (horizontal line) and the first and third quartile. An 'a' on top of the box plots indicate which combinations (left) or which treatments (right) are not significantly different from each other (two-tailed t-test, $p < 0.05$).

## RESULTS

Addition of AZ reduced growth of the foraminifera (Fig. 2). Without AZ, specimens grew approximately 8 times larger than the foraminifera incubated with the AZ. Within each of the two treatments, there was no consistent trend in growth (expressed as area of the foraminiferal shell photo's) with $pCO_2$. For the control groups (*i.e.* no AZ added), only those cultured at highest $pCO_2$ were significantly larger (t-test, $p$-value > 0.05) than those at lower CO₂'s, while those grown at $pCO_2$'s of 400–1,200 were not significantly different from each other (t-test, $p$-value > 0.05). For the foraminifera that were incubated with AZ, only those cultured at a $pCO_2$ of 400 ppm were significantly smaller (t-test, $p$-value < 0.05) than those incubated at the other $pCO_2$'s. The sizes of the foraminifera from 800, 1,200 and 1,600 ppm CO₂ were not significantly different from each other (t-test, $p$-value > 0.05).

Element incorporation was also significantly different in the experiments in which AZ was added. Foraminifera grown with AZ have 3 times more Mg in their shell than the specimens grown without AZ (Fig. 3). For Na/Ca, this effect was smaller, with twice as high ratios with than without the inhibitor added. Sr/Ca was least affected by presence of AZ, with about 10% higher Sr/Ca values in case AZ was added. The effect of $pCO_2$ on element incorporation was absent for all three El/Ca when AZ was added. In the presence of AZ, only Mg/Ca increases linearly with $pCO_2$ (OLS regression, $p < 0.05$).

Indicated are the Mg/Ca (top), Na/Ca (middle) and Sr/Ca (lower panel) of newly grown chambers from four different $pCO_2$s. Specimens cultured without addition of AZ are in blue, with addition of AZ in orange. Box-whisker plots indicate the average ('X'), the median (horizontal line) and the first and third quartile. Dots indicate individual

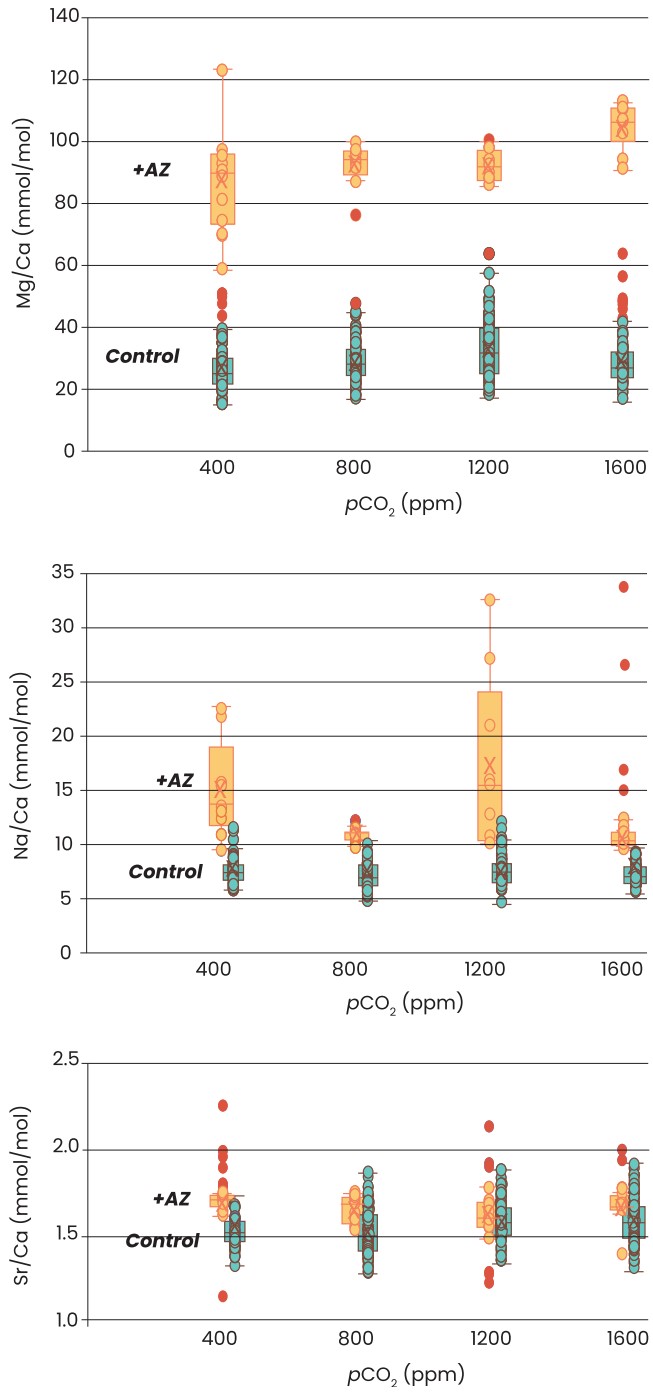

**Figure 3 Elemental ratios as a function of CO₂.** Recorded size of the foraminifera cultured at different $p$CO$_2$'s is plotted without (left) and with addition of AZ (right): note the difference in the scale of the y-axis. Dots indicate individual foraminifera: outliers are indicated in red and are lower than 1.5 times the lower quartile or higher than 1.5 times the third quartile. Box-whisker plots indicate the average ('X'), the median (horizontal line) and the first and third quartile. An 'a' on top of the box plots indicate which combinations (left) or which treatments (right) are not significantly different from each other (two-tailed t-test, $p < 0.05$).

foraminifera: outliers are indicated in red and are lower than 1.5 times the lower quartile or higher than 1.5 times the third quartile. The two treatments are horizontally shifted in the lower two panels to show the full scale of variability in each of them.

## DISCUSSION

### Growth rate is affected by AZ and by pCO$_2$

The clear effect of AZ on growth in *Amphistegina lessonii* (Fig. 2) indicates that conversion between bicarbonate and carbonate by Carbonic Anhydrase (CA) is somehow necessary for growth by foraminifera. This confirms an earlier culturing experiment with foraminifera (*De Goeyse et al., 2021*) and is in line with the importance of CA in calcification in other marine organisms (*e.g.*, corals: *Moya et al., 2008*; bivalves: *Cardoso et al., 2019*). Earlier work on the role of CA on calcification in *Amphistegina* showed that addition of this enzyme reduces calcification (and stimulates photosynthesis; *Ter Kuile, Erez & Padan, 1989*). This led to the conclusion that CA harms calcification, for example by reducing the content of intracellular carbon pools. Results in the same work showed, however, that calcification was also negatively impacted after addition of the CA-inhibitor ethoxyzolamide, indicating that both reduced and increased CA activity are detrimental for inorganic carbon uptake.

The involvement of CA in foraminiferal calcification indicates that they do not (primarily) rely on the available (CO$_3^{2-}$) in the surrounding seawater. The uptake of bicarbonate and/ or CO$_2$ and subsequent conversion into CO$_3^{2-}$ at the site of calcification is consistent with biomineralization models that highlighted pH regulation (*e.g.*, *De Nooijer et al., 2009*; *Toyofuku et al., 2017*). Fluorescent labelling showed that during calcification, protons are pumped out of the site of calcification and into the surrounding seawater (*De Nooijer, Toyofuku & Kitazato, 2009*; *Glas, Langer & Keul, 2012*; *Toyofuku et al., 2017*), shifting the inorganic carbon speciation towards CO$_2$ outside the foraminifera and towards CO$_3^{2-}$ inside the foraminifera, which facilitates inward CO$_2$ diffusion. Both inside and outside the foraminifer, conversion between inorganic carbon species would benefit from the presence of CA (Fig. 4).

The conversion of CO$_2$ into HCO$_3^-$ and vice versa is approximately 10$^7$ times faster in the presence of CA than in its absence (*Zeebe & Wolf-Gladrow, 2001*; *Bertucci et al., 2013*). The conversion of bicarbonate to CO$_2$ is particularly slow (with kinetics in the ~15 s range) and therefore the aid of CA in this reaction is largest. If foraminifera rely on diffusion of CO$_2$ from the seawater into the site of calcification, the conversion outside the foraminifers (HCO$_3^-$ -> CO$_2$) could particularly benefit from the catalysis by CA. Since AZ is membrane-permeable (*Supuran & Scozzafava, 2004*), the effect of AZ could also indicate the presence of CA inside the cell, at the site of calcification, where it may help the conversion of CO$_2$ into CO$_3^{2-}$ (Fig. 4).

When rotaliid foraminifera calcify, a thin layer of cell material (the protective envelope) separates the calcifying fluid from the surrounding seawater. Within the site of calcification, conditions are manipulated by transport of ions to promote calcium carbonate precipitation (left side). In addition, ions are also transported into the calcifying fluid by vacuolization of seawater (right). The exact ion transport mechanisms, the balance

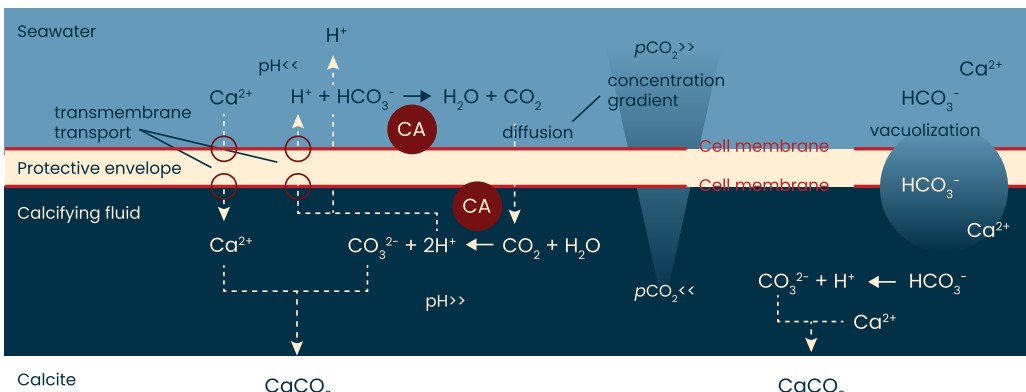

**Figure 4 A simplified model of calcification in foraminifera.** When rotaliid foraminifera calcify, a thin layer of cell material (the protective envelope) separates the calcifying fluid from the surrounding seawater. Within the site of calcification, conditions are manipulated by transport of ions to promote calcium carbonate precipitation. The exact ion transport mechanisms, the volume of the calcifying fluid, as well as the thickness of the protective envelope remain to be identified and quantified, respectively. Previous work has shown, however, that during calcification protons are pumped from the calcifying fluid to the surrounding seawater. The resulting pH gradient (high inside the calcifying fluid, low outside the foraminifer) will create a $p$CO$_2$ gradient that will remain as long as the indicated reactions proceed. Indicated in red are the hypothesized locations of the enzyme carbonic anhydrase (CA): one near the outer cell membrane of the protective envelope and one near the inner membrane. Schematic based on *Erez (2003)*, *Toyofuku et al. (2017)* and *Pacho et al. (2024)*.

between transmembrane transport and vacuolization, the volume of the calcifying fluid, as well as the thickness of the protective envelope remain to be identified and quantified, respectively. The protons that are pumped from the calcifying fluid to the surrounding seawater result in a pH gradient (high inside the calcifying fluid, low outside the foraminifer) and hence leads to a $p$CO$_2$ gradient that will remain as long as the indicated reactions proceed. Presence of symbionts and contribution from respiration may well affect the pH gradient and thus affect the diffusion of CO$_2$ into the site of calcification. Indicated in red are the hypothesized locations of the enzyme carbonic anhydrase (CA): one near the outer cell membrane of the protective envelope and one near the inner membrane. Schematic based on *Erez (2003)*, *Toyofuku et al. (2017)* and *Pacho et al. (2024)*.

The results from our controlled growth experiments show that there is a small, but positive effect of CO$_2$ on growth in *Amphistegina lessonii* (Fig. 2), which is in line with earlier experiments (*Dämmer et al., 2023*). Either the lowest set CO$_2$ conditions resulted in smaller specimens (with AZ) or the highest CO$_2$ resulted in specimens growing larger shells (without AZ). This suggests that foraminifera do not rely on (CO$_3^{2-}$) itself, since this would have resulted in a decrease with increasing $p$CO$_2$ values (as in sea urchin larvae: *Matt, Chang & Hu, 2022*). The observed positive effect of $p$CO$_2$ on calcification is consistent with the active regulation of pH at the site of calcification by foraminifera. The implication of such a mechanism is that an increase in the total dissolved inorganic carbon (DIC) availability, regardless of the ratio between inorganic carbon species, potentially enhances calcification by increasing the DIC pool available to be actively converted into (CO$_3^{2-}$). Such a positive effect of (DIC) or $p$CO$_2$ on calcification in foraminifera has been shown before (*e.g.*, *Hikami et al., 2011*; *Fujita et al., 2011*; *Vogel & Uthicke, 2012*; *Dämmer*

*et al., 2023*), although other studies also showed a decrease in calcification rates at elevated $CO_2$ levels (*Allison et al., 2010*; *Barker & Elderfield, 2002*; *Bijma, Spero & Lea, 1999*; *Bijma, Hönisch & Zeebe, 2002*; *Dissard et al., 2010*; *Gonzalez-Mora, Sierro & Flores, 2008*; *Keul et al., 2013*; *De Moel et al., 2009*; *Oron et al., 2020*).

The observed discrepancy may reflect the ability or degree to which extent a species is capable to regulate their internal pH. Strong *vs* weak pH regulation has also been hypothesized to explain variability among marine metazoa to cope with ocean acidification (*Ries, 2011*). Within foraminifera, pH regulation likely also varies between species (*De Nooijer et al., 2023*), explaining the variability in the responses to elevated $CO_2$. An analysis of the presence and location of carbonic anhydrase among foraminiferal species could help to better understand the interplay between pH regulation, inorganic carbon utilization, photosynthesis and shell chemistry.

## Impact of AZ and $CO_2$ on Mg/Ca, Na/Ca and Sr/Ca

The interplay between calcification and CA is evident from the large impact on the Mg/Ca of newly precipitated calcite after addition of AZ (Fig. 3). When AZ is added the relatively limited amount of carbonate precipitated has much higher Mg/Ca values. The fact that AZ slows down growth (Fig. 2) implies that the increased Mg/Ca ratios are not caused by higher precipitation rates (as shown in inorganic precipitation experiments; *Burton & Walter, 1991*; *Mavromatis et al., 2015*). Although growth by a foraminifer is not necessarily the best predictor for calcite crystal's precipitation rate, it is unlikely that slow rates of chamber addition (Fig. 2) and thin foraminiferal chamber walls are accomplished by higher precipitation rates. Instead, the hampered uptake of inorganic carbon by the inhibition of CA by the added AZ indicates a close coupling between the uptake of the cat- and anions necessary for calcification, assuming an equivalent proportion of high- and low-Mg/Ca bands within the shell walls. With CA being inhibited, affecting bicarbonate-carbonate ion conversions, either Mg uptake is enhanced or Ca intake reduced: the heterogenous distribution of Mg (and other ions) within the shell's wall may provide an explanation for the elevated Mg/Ca at reduced CA activity.

Incorporated elements like S, Ba, Mg, Na, *etc.* within foraminiferal shell walls have been shown to alternate between high- and low-concentration bands (*Kunioka et al., 2006*; *Spero et al., 2015*; *Fehrenbacher et al., 2017*; *Geerken et al., 2019*). Elevated Mg/Ca, S/Ca, *etc.* often coincide with the first layers precipitated, which can be close to the primary organic layer of a new chamber or the layer formed on top of a pre-exiting chamber (*Tyszka et al., 2021*), as is typical for rotaliid species (*Reiss, 1957*; *Erez, 2003*). In the planktonic *Orbulina universa* (*Spero et al., 2015*) and *Neogloboquadrina dutertrei* (*Fehrenbacher et al., 2017*), this high/low banding is shown to be modulated by day/night cyclicity. Irrespective of the environmental trigger(s), incorporated magnesium, strontium, *etc.* and their layered occurrence are likely resulting from a combination of two different proposed modes of calcification: seawater transport and selective, transmembrane ion transport.

*Toyofuku et al. (2017)* hypothesized that in low-Mg/Ca rotaliid foraminifera, the observed outward proton pumping (from the site of calcification; *De Nooijer et al. (2009)* to the surrounding seawater; *Glas, Langer & Keul, 2012*) is proportional to an inward $Ca^{2+}$
flux (*Nehrke et al., 2013*) and hence the flux of DIC from the seawater to the site of calcification. The inward diffusion of $CO_2$, due to the steep site of calcification-seawater pH gradient (approximately 9 to 6; *De Nooijer et al. (2009)* and *Glas, Langer & Keul (2012)*, respectively), provides the necessary protons to maintain the exchange of two protons (out) for one calcium ion (in). It should be noted that the site of calcification likely has a very small volume (*e.g.*, *Nagai et al., 2018*) so that fluxes to the site of calcification will lead to high levels of oversaturation. In addition, it may be that it enters the site of calcification by the $Ca^{2+}$ channel or -pump (*De Nooijer et al., 2014*) because of the Ca pump not being able to discriminate perfectly against Mg (see *Dubicka et al., 2023* for visualization of $Ca^{2+}$ and $Mg^{2+}$ within foraminiferal cells).

Alternatively, seawater vacuolization may supply ions like $Mg^{2+}$ to the site of calcification (*Erez, 2003*; *Bentov & Erez, 2006*; *Bentov, Brownlee & Erez, 2009*). For *Amphistegina*, studies have suggested a relatively large internal ion pool, fueled by seawater uptake prior to calcification (*Ter Kuile & Erez, 1988*). This may mean that the large internal-external pH (and thereby $p$CO$_2$) gradient is smaller in *Amphistegina* than described before (*De Nooijer et al., 2009*; *Glas, Langer & Keul, 2012*). It may well be that both modes of ion supply are responsible for the average El/Ca, as well as their layered pattern. The start of chamber formation is likely accompanied by enclosure of seawater, containing $Mg^{2+}$, $Na^+$, *etc.* that are incorporated during chamber formation (*Geerken et al., 2022*). Incorporation of these elements, plus the subsequent dilution by selective $Ca^{2+}$-pumping will lower the ratio of (*e.g.*,) $Mg^{2+}$ relative to $Ca^{2+}$ (and thus the Mg/Ca in the calcifying fluid) and result in a lower-concentration band formed after the first high-concentration band is formed. It has also been hypothesized that $Mg^{2+}$ is selectively removed from the site of calcification (*Bentov & Erez, 2006*), but indications for that are currently lacking. For $Sr^{2+}$, it is likely that some is co-transported through these Ca-pumps (*Hagiwara & Byerly, 1981*; *Baraibar et al., 2018*). With a disturbance of the calcification process due to inactive CA, walls may become significantly thinner and thus the contribution of this first layer will be larger. This will elevate the average Mg/Ca, but will have less effect on Na/Ca and Sr/Ca (Fig. 3). In general, the banding observed for these latter two elements is less pronounced (Na/Ca; *Bonnin et al., 2016*; *Van Dijk et al., 2019*) or absent (Sr/Ca; *Kunioka et al., 2006*; *Geerken et al., 2019*) and thinner shell walls will thus have less impact on their average concentrations.

Differences in how well-pronounced these bands within the shell wall are, is determined by element-specific inorganic partition coefficients (*e.g.*, *Tesoriero & Pankow, 1996*) and subsequent Rayleigh fractionation during continued calcification (*Elderfield, Bertram & Erez, 1996*). Ions that are easily incorporated into the calcite crystal lattice (*i.e.*, described by a high partition coefficient) will decrease relatively rapidly in the fluid at the site of calcification, compared to ions that fit poorly in calcite. Together with the fraction of the ions remaining in the calcifying fluid (*i.e.*, Rayleigh fractionation; *Elderfield, Bertram & Erez, 1996*; *Evans, Müller & Erez, 2018*), these processes determine the average El/Ca in the precipitated shell wall.

The experiments performed here, in which calcification is blocked, can serve to test the Rayleigh fractionation model for element incorporation in foraminiferal shell calcite,

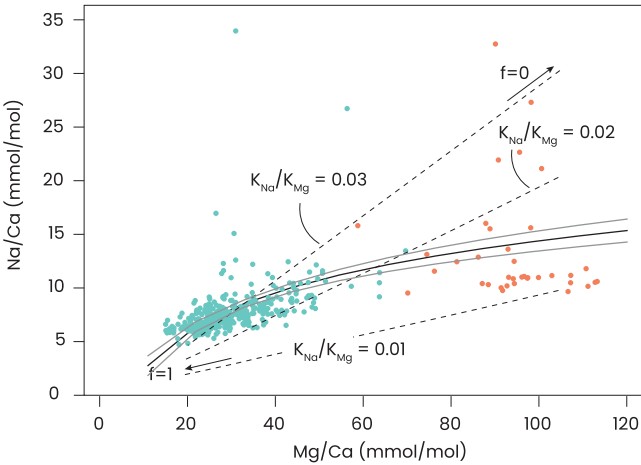

**Figure 5 Relation between incorporated Mg and Na.** Na/Ca *vs* Mg/Ca in foraminifera grown under control conditions are shown in green and with AZ added in orange. A regression described by Na/Ca = 5.2(±0.39) * e^Mg/Ca – 9.7(±1.4) is plotted as a black line with the 95% confidence interval of this regression in grey. Added are dashed lines indicating relations for some combinations of inorganic Mg- and Na-partition coefficients ($K_{Mg}$ and $K_{Na}$), which are plotted as a function of f (fraction of Ca remaining in the calcifying fluid). The right-hand side of the dashed lines indicate f = 0 (no Ca remains) and the left-hand side f = 1 (almost no $CaCO_3$ precipitates).

assuming that the inorganic carbon uptake and calcium pumping are directly coupled. For the inorganic partition coefficients we assumed 0.015–0.020 for magnesium (*Oomori et al., 1987*; *Mucci, 1987*) and 0.0015–0.0025 for Na (*Busenburg & Plummer, 1985*; *DeVriendt et al., 2021*). The relations for the simplified Rayleigh distillation model (Eqs. (1) and (2)) shown in Fig. 4 indicate that the correlation between foraminiferal Mg/Ca and Na/Ca in our dataset can be approximated with a simple Rayleigh fractionation process. Differences between the model outcomes and foraminiferal data (Fig. 4) can be explained by additional controls during biomineralization that may modulate the Mg- and/or Na-partitioning. In particular, a linear trendline through the data would have a non-zero intercept and may indicate that at high values for f, other processes that affect Mg- and Mg-partitioning differently, play a larger role. Such processes may include phase transformations, selective ion transport, precipitation rates, *etc*.

The relatively high spread in El/Ca values in the AZ-treated specimens is likely related to the somewhat larger uncertainty in the analyses caused by the thin shell walls because of the limited amount of carbonate added by the foraminifera under these treatments. Elements in foraminiferal calcite may well be affected by a inward flux of Ca, rather than precipitating from a seawater-like solution. Alternatively, the correlation between El/Ca (Fig. 5) may be explained by distortion of the lattice-strain due to the incorporation of an ion, promoting the incorporation of another one. *Mucci & Morse (1983)* have shown that such an explanation applies to Mg- and Sr-incorporation in inorganically precipitated calcites, while *Evans et al. (2015)* showed the same for Mg- and Na-incorporation. These two hypotheses (Ca-dilution and lattice-strain balancing) do not exclude each other and

future studies will have to show to what magnitude they can explain the correlation between concentrations of elements in foraminiferal calcite.

The effect of the inorganic carbon system on foraminiferal element incorporation showed a negative effect of ($CO_3^{2-}$) on Mg/Ca (*Dissard et al., 2010*; *Allen et al., 2016*). An experiment with *Amphistegina lobifera*, on the other hand, showed an increase in Mg/Ca with decreasing pH (*Levi, Müller & Erez, 2019*), which is in line with our results (Fig. 3) and results from studies on planktonic species (*Kisakürek et al., 2008*; *Russell et al., 2004*; *Lea, Mashiotta & Spero, 1999*; *Gray & Evans, 2019*). Experiments with several large benthic foraminiferal species, showed no effect of $p$CO$_2$ on Mg/Ca, Na/Ca and Sr/Ca (*Van Dijk, De Nooijer & Reichart, 2017*). The combination of these reports and the results shown here suggest that the minor effect of pH/CO$_2$/($CO_3^{2-}$) on Mg incorporation is likely an indirect effect, possibly related to variability in $Ca^{2+}$ pumping rates or the ratio of high- and low-Mg/Ca bands. This underscores the necessity to unravel the foraminiferal calcification pathway when aiming to explain the effect of environmental variables on their calcite's composition.

## CONCLUSIONS

Calcification in the benthic foraminifer *Amphistegina lessonii* critically relies on carbonic anhydrase. With the inhibitor acetazolamide added to the culturing medium, sizes of the specimens were 5–10 times smaller than in the control group. The involvement of carbonic anhydrase fits the conversion of bicarbonate into $CO_2$ and vice versa, as hypothesized in recent models for calcification in foraminifera. The steep increase in the calcite's Mg over Ca concentration after addition of acetazolamide further suggests that Ca-pumping and inorganic carbon uptake are directly coupled and has a major impact on El/Ca values. The positive (although relatively small) effect of increased ambient $CO_2$ on calcification confirms that foraminiferal pH regulation decouples calcification from the saturation state in the surrounding seawater.

### Funding
This work was supported by the Dutch Research Council (ALWOP210). The funders had no role in study design, data collection and analysis, decision to publish, or preparation of the manuscript.

### Grant Disclosures
The following grant information was disclosed by the authors:
Dutch Research Council: ALWOP210.

### Competing Interests
The authors declare that they have no competing interests.
## Author Contributions

- Siham De Goeyse conceived and designed the experiments, performed the experiments, analyzed the data, prepared figures and/or tables, authored or reviewed drafts of the article, and approved the final draft.
- Chiara Lesuis performed the experiments, analyzed the data, authored or reviewed drafts of the article, and approved the final draft.
- Gert-Jan Reichart analyzed the data, authored or reviewed drafts of the article, and approved the final draft.
- Lennart de Nooijer conceived and designed the experiments, performed the experiments, analyzed the data, prepared figures and/or tables, authored or reviewed drafts of the article, and approved the final draft.

## Data Availability

The inorganic carbon chemistry (DIC and TA), as well as the growth and foraminiferal geochemistry, are available in the Supplemental File.

## Supplemental Information

Supplemental information for this article can be found online at http://dx.doi.org/10.7717/peerj.18458#supplemental-information.

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
