# Peer review of "The effect of carbonic anhydrase on foraminiferal Mg/Ca"

_PeerJ, doi:10.7717/peerj.18458_

## Round 0.1 · original submission · Major Revisions

· Academic Editor

Major Revisions

We have now received three detailed reviews of your manuscript. All reviewers comment positively on your work, but also recommend revisions. They point to the need to clarify some of the methods, and all three suggest substantial revisions to the discussion.

Please revise your manuscript based on the reviewers’ constructive feedback and prepare a detailed response to the individual points raised, outlining your revisions in the manuscript or reasoning why you disagree with a point raised.

I look forward to receiving your revised manuscript.

Reviewer 1 ·

Basic reporting

This study reports an enzyme control on the calcification of benthic foraminifera Amphistegina lessonii. The data provide better understanding to mechanisms of foraminiferal shell formation, and potentially predict how tolerate marine calcifiers can be under a high CO2 world. The results are exciting, evident and valuable to the biomineralisation community. However, I cannot recommend this article for publication in its current state. My main concern is the biomineralisation model in the discussion (L339-380) is not supported by sufficient details, which potentially leads to an over assertive conclusion. I outline my suggestions as follows:

1. I agree that whether the element incorporation is affected by AZ has correlation with its banding pattern in shells. However, a lack of background information and mixing of hypotheses make it difficult to follow the argument. I suggest separate paragraphs introducing (1) El/Ca banding in foraminifera, (2) various biomineralisation models (the H+/Ca2+ pumping and the seawater vacuolization), and then combine the two ideas (banding and pumping) to discuss the data in this study.

L344-354 does not necessary explain the difference between Mg/Ca, Na/Ca, and Sr/Ca shown in Fig. 3. If the low Mg band is formed due to dilution of inward Ca pumping as suggested by the authors, why does the other elements, such as Na and Sr, not diluted the same way and form a low element band?

L354-356 is confusing. More explanations linking the banding patterns to the data in this study are needed.

L361-362 is too assertive under the context, as the discussion assumes that foraminifera calcification is limited by CO2 inward diffusion and following conversion into HCO3-. This does not necessarily mean active pumping being blocked.

L364: Isn’t the temperature constant during the incubation? If that is the case, why is temperature a possible environmental impact here?

2. In L357-369, I suggest testing data with Rayleigh distillation or other models that may apply. I hope it will produce stronger evidence to any possible biomineralisation mechanisms supported by the data.

3. To support the statement in L376-378, a correlation (such as Fig 4.) on Sr and Mg need to be shown as well.

Although the results are important and exciting, the discussion section needs more organization and thoughts to support the authors’ statements. If the authors address these concerns, along with my other comments below, I would be excited to recommend this article for publication. I would be happy to review a second time if necessary.

Experimental design

The experiment is well designed for the research questions, and carefully controlled throughout the 5-week incubation period. However, more details are needed for the three sections listed below.

1. References are needed for measurements of alkalinity, DIC, and nutrient analysis (L155-L163). Please provide the typical analytical errors. I suggest remove ‘(Fig. 1)’ in L160, as I would then expect Fig.1 to be a diagram showing experiment procedures in such case. What nutrient parameters were measured? The nutrient results should be reported if mentioned here.

2. The content in the statistical analysis part (L210-220) need more details. The results of the (dis)similarity between replicates mentioned in L210-211 is not reported in the manuscript. Although I believe the authors must have confirmed that the replicates are identical to each other, it is a necessity to show the results in either a plot or a table. I suggest editing the L211 into ‘…for all replicates of each condition in the control group’, since, if I understand correctly, only the control group have all 3 replicates analyzed. I also suggest adding statistical test between different pCO2 conditions in L240-245 into this section. It is much clearer that the variations between (1) replicates and (2) different pCO2 conditions are both considered and treated statistically.

3. Please provide the results of comparison between solution and LA-ICP-MS data (L222-233) in a plot or a table.

Other specific notes:
L114: What is the porosity of the filters used to filter seawater?
L136: Add ‘dimethyl sulfoxide’ in front of the initial DMSO.
L169: Please provide recipe of 0.33% H2O2 solution in 0.5M NH4OH (e.g., 1:1 v/v)
L178-180: Is there a specific reason why (1) only 10 out of 30-40 individuals per flask, and (2) only one replicate in the AZ group were measured?
L201: Please cite the Fiji: doi:10.1038/nmeth.2019
L227: Add description such as ‘trace elements to calcium ratio in the shells’ in front of the initial ‘El/Ca’.

Validity of the findings

The effect of the enzyme inhibitor on the foraminiferal calcification is unambiguous and exciting. However, the influence of pCO2 is not that evident. The authors state that CO2 has small but positive effect on growth rate and magnesium partitioning into calcite shells, which are not that obvious from the plots (Fig.2 &3). I suggest do a linear regression check between each x and y pairs to support the statement.

Additional comments

For the figures:
1. I suggest increase the font size as the text is in general too small to read easily.
2. ‘Supplementary fig. 1’ mentioned in L133 does not exist.
3. In figure 3, I suggest putting the box plots of the two groups (Control v.s. +AZ) alongside with each other instead of on the same line. It will be easier to identify the outliers of each group. The scale of y-axis in the lower panel (Sr/Ca) can be adjusted to 1.0 to 2.5.
4. The title of figure 4 ‘Comparison of solution and laser-ablation-ICP-MS’ does not match the content. Words like ‘are shown’ and ‘and’ in the description are weirdly in bold. Maybe the 95% confidence interval can be colored in semi-transparent grey, shown as enveloping shaded area around the regression line.

For the raw data in supplemental files:
1. Please provide units.

Other specific notes:
L22: foraminifers -> foraminifera
L122&131: tissue flasks -> tissue culture flasks
L178: and -> per
L257-259: The two sentences seem repeating each other.
L303: [CO2] -> pCO2
L349-350: I would prefer the term ‘organic layers’ instead of ‘organic lining’, as the latter can also be referred to as the pseudochitinous remains inside foraminiferal shells (see Tyszka et al., 2021. dio: 10.1016/j.earscirev.2021.103726).
L356: Please provide references for Na/Ca and Sr/Ca bandings.

Cite this review as

Reviewer 2 ·

Basic reporting

Some methodological details missing, please see below.

Experimental design

No comment.

Validity of the findings

Please see my additional comments.

Additional comments

The manuscript ‘The effect of carbonic anhydrase on foraminiferal Mg/Ca’ by De Goeyse et al. reports growth rate and trace/major element data for a species of symbiont-bearing benthic foraminifera cultured under four different pCO2 and with/without the presence of a carbonic anhydrase inhibitor (acetazolamide; AZ) in the growth medium. The authors find that the addition of AZ results in a strong decrease in growth, whereas pCO2/pH exerts, if anything, a minor control. Shell geochemistry measurements demonstrate that foraminifera grown in the presence of AZ are characterized by very different chamber element/Ca ratios, with Na, Mg, and Sr all present at higher concentrations in the AZ group. Again, in contrast, pCO2/pH has no/little effect.
I really enjoyed thinking about the results of this interesting study, which is well designed to answer the key questions posed here. The authors present a large amount of data as well as some interesting hypotheses to explain their results, specifically that i) CA facilitates the conversion of HCO3- to CO2, enabling more efficient diffusion to the calcification site, and ii) that inhibiting CA affects shell geochemistry as the foraminifera switch to dominantly precipitating primary calcite rather than secondary layers, which are known to have a different composition.
My main suggestion for improvement is that, as interesting as these results are, the explanation that the authors put forward misses key published evidence and cannot explain the observation that culturing a very similar foraminifer in the presence of CA also strongly inhibits calcification. In my view, the discussion is also insufficiently balanced in terms of placing these results in the context of what we understand about the biomineralisation of this genus, and some key methodological details are missing from the manuscript. Otherwise, I congratulate the authors on a nice study and look forward to seeing it published once these issues have been addressed.

Main comments
1. The missing discussion point is that the opposite experiment has been previously performed. ter Kuile et al. (1989) cultured Amphistegina lobifera in the presence of CA and demonstrated that doing so also greatly inhibits calcification, which is, in effect, the opposite of the results presented here that argue that CA is an essential part of calcification. Those authors hypothesized that the presence of CA results in a collapse of the internal carbon pool driven by CO2 diffusion out of this pool catalyzed by the presence of CA (note that contrary to what is written in the present manuscript, Amphistegina has a large and important internal C pool that is involved in calcification; see papers by ter Kuile). This doesn’t diminish the importance of the results presented here, but it does mean that the discussion needs substantial revision, and the key hypotheses presented here need to be removed or revised to account for all key observations including this. Also note that ter Kuile et al. (1989) could not find measurable amounts of CA in these foraminifera.
2. The discussion appears to conflate the effect of CA inhibition with that of (possible) Ca transport (e.g. in the abstract, lines 30-34). What we can say from the data presented here is that AZ strongly inhibits calcification but I cannot find any evidence in the data to suggest that this has any impact on ion transport processes. The authors argue that the differences in shell geochemistry in cultures performed in the presence of AZ can be explained by a shift towards chambers dominated by primary calcite, which sounds reasonable to me. Given this, I don’t see the need to invoke an additional effect of AZ on ion transport processes. And at the very least, the evidence for this needs to be presented much more clearly.
3. We can (and do!) continue to debate the importance of ion transport versus seawater vacuolization and I don’t think every argument for and against the two hypotheses needs to be repeated here. However, this is the genus for which we have many published observations of the importance of seawater vacuolization, such that I found the discussion too imbalanced in terms of presenting the data within the framework of the two competing hypotheses (the word ‘vacuole’ does not appear in the manuscript!). Given that the geochemical data do not in themselves argue for the authors’ preferred hypothesis (see #2), I’m not sure this issue needs to be debated here at all – why not simply talk about the mechanism of carbon transport to the calcification site – but if it is to be included, then the discussion should include the many papers and arguments that have demonstrated the importance of seawater transport in this genus.
4. The experiment is overall well described, but there appears to be something missing or a mistake. The authors mention solution ICPMS measurements but do not present them or mention any methodological/analytical details, while Fig. 4 clearly does not show solution data. Please correct and describe the solution analyses properly if they are to be included.

Minor comments
1. Lines 50-51. You could also mention that most species do not have the same Mg/Ca-T slope as inorganic calcite.
2. Lies 75-76. It might involve Ca2+ transport but it might equally involve e.g. a Na+-proton pump. The requirement for pH elevation and carbon concentration does not need to have any implications for Ca transport. From this, the point on lines 78-79 needs caveating at the very least (and even if a Ca pump is involved, how much would we expect the calcification site [Ca2+] to change given a Ca/proton ratio of ~106 in seawater?).
3. Lines 86-89. It may be high at the calcification site but it is low in the cytosol. Is the calcification site in direct contact with the boundary layer in this species?
4. Line 128. Please state what this uncertainty represents.
5. Section starting on line 154. Please give estimates of data quality for these techniques. In addition, were any TAlk measurements of the culture jars made prior to the water changes? To what extent did the foraminifera alter their jar carbonate chemistry?
6. Lines 174-176 and 229-231. In an earlier methods section you state that foraminifera were transferred to culture immediately after reproduction. Why is there the need to target only the final chambers if there is therefore presumably almost no pre-culture calcite? Why would this be a cause of offset between solution and LA?
7. Line 181 onwards. Some basic details are missing, for example which instruments were used, laser wavelength, the tuning conditions, gas flow rates etc.
8. Line 186. Used to correct for drift in what way? Please fully detail the data processing.
9. Line 200. I think you mean cross sectional area.
10. Section starting on line 221. If you would like to include these results then please detail the sample preparation and analytical methodology properly. In addition: please quantify ‘slightly’ (line 226), and why do you quote 1SD (line 228)? Surely we’re interested in the differences between the mean values?
11. Line 262. Please quantify ‘small’. Was it significant?
12. Line 276. Consider replacing ‘calcification’ with ‘growth’.
13. Line 320. You could cite Oron et al. (2020) here, relevant because symbiont-bearing benthic is more similar to the species utilized here than in many of the studies you do cite.
14. Line 340. This statement is much too general. It may not be the case for all species, but Amphistegina have a large internal carbon pool that is used in calcification. Indeed, ter Kuile & Erez (1988) showed that calcification only begins once the pool is filled.
15. Line 345. Can you cite an example of a Ca2+ channel or pump that may be involved in pH regulation but does not discriminate strongly against Mg2+?
16. Discussion starting on line 339 and line 404. I don’t understand the need to invoke Ca or Mg transport at all here. Could it not simply be that AZ inhibits the formation of secondary calcite without exerting any control on ion transport itself (see my main comment above).
17. Line 362. On a similar note. Why would we imagine that these experiments block active pumping? They may simply inhibit calcification (via the organisms ability to concentrate C) without having any impact on active transport.
18. Line 370. A very simple alternate explanation for the data is that Mg incorporation into calcite is known to have a substantial impact on the incorporation of other elements via lattice distortion. Mucci & Morse (1983) demonstrated this for Sr, and Evans et al. (2015) suggested it is also likely to be the case for Na. The slopes from those two studies coupled with the difference in Mg/Ca between the control and AZ groups suggests that all or most of the difference in mean values can be explained in this way (e.g., the slope of your Fig. 4 is almost identical to the slope in Fig. 7 of Evans et al.). Note that this also explains your Sr data. From there, the following discussion is therefore also likely incorrect and the alternative hypothesis should at least also be given.
19. Lines 387-388. Most studies on planktonic foraminifer have shown a negative relationship (G. ruber, O. universa, G. bulloides), e.g. Kisakürek et al. (2008), Russell et al. (2004), Lea et al. (1999), Gray & Evans (2019), in line with Levi et al. (2019).
20. Figure 4. See main comment above.

Typos
- Line 159. Capital L in HgCl2.
- Line 173. Measured.
- Figure 2 & 3 caption. No apostrophe needed after CO2.

Cite this review as

Reviewer 3 ·

Basic reporting

This work by De Goeyse et al. investigated the effect of the carbonic anhydrase inhibitor Azetazolamide (AZM) on the calcification rate and elemental compsition of the shell in the large benthic foraminifera A. lessonii. In addition, foraminifera were exposed to different CO2-induced acidification levels to study the combined effects of CA inhibition and hypercapnic stress. The study demonstrated strong reductions in calcification rates in the presence of AZM accompanied by an enhanced incorporation of Mg2+ or a reduced incorporation of Ca2+ into the mineral. Only minor effects were observed by the different pCO2 regimens with slightly increased calcification rates under high pCO2 conditions. This is a small but interesting dataset on the role of CA in the mineralization process of a foraminifera species under different pCO2 regimens. Since this manuscript is rather descriptive the mechanistic explanations for the observed effects are highly speculative. I strongly suggest to revise the manuscript (in particular introduction + discussion) in a way that the discussion refers more closely to the data presented rather than being too speculative. A strong focus of the text is dedicated to the compartments and their proton gradients that need to be much more precisely characterized to allow any hypothesis regarding CO2 gradients.

Experimental design

Methods:
L: 176-177: Is it possible that no chamber was produced at all in the presence of AZM? This would lead to strong misinterpretations of the results from the LA-ICP-MS analyses. It would be very helpful if the authors could provide a prove that the regions alalyzed by LA-ICP-MS were generated under the presence of AZM (e.g. calcein labeling).

Validity of the findings

1. There are several sections in this manuscript that describe the pH regulation in diverse cellular and extracellular compartments which are extremely vague, and may lead to misinterpretations. To my knowledge foraminifera are intracellular calcifiers that have alkaline vesicles that potentially generate an amorphous CaCO3 phase that is transported and exocytosed towards the calcification front. In their manuscript the authors refer to pH gradients between the site of calcification (SOC) and outside the SOC. This is extremely vague as this can be between the lumen of vesicles and the cytosol, or between the cytosol and the outside of the foraminifera or between the lumen of vesicles and the seawater. Also, the bafilomycin sensitive transport of protons has been observed on the surface of foraminifera but not between the calcification compartment and the external medium.
These pH gradients need to be precisely described for the proper interpretation of the discussion on CO2 enhanced calcification.

2. I have a serious problem with the proposed inard flux of CO2 by pH gradients. Maybe Co2 generated by the mitochondria within the cell can be used to fuel photosynthesis by the endosymbionts, but I cannot imagine an influx of CO2 from the seawater into the cell by the pH gradients present in these organisms. At an extracellular pH of approx 8 as well as an intracellular pH of approx.. 7 the majority of DIC is in the form of HCO3-. Only extreme acidification pH < 5 in the boundary layer may cause inward directed CO2 flux assuming a typical pCO2 for aquatic cells (Melzner et al. 2009 Biogeosciences 6:2313-2331). As has been demonstrated in marine algae and the sea urchin larva (Thoms et al. 2001 J. Theor. Biol. 208, 295–313; Trimborn et al. Physiol. Plant. 133, 92–105; Price & Badger 1989 Plant Physiol. 91, 505–513; Matt et al. 2022 PNAS 4:119), an uptake of HCO3- in concert with extracellular carbonic anhydrase may be more likely.


Introduction:
L 86: „…inward diffusion of CO2 as a result of the steep pH gradient (high inside, low outside the foraminifer)…..“ What is „inside“ the foraminifer? The cytosol or the alkaline vesicles? If the authors refer to the alkaline compartments, they need to explain how a typical intracellular pH of 7 should favour the influx of CO2.
L93 „also help to convert the Co2 to CO32-…“ CA catalizes the reversible hydration of CO2 to form the H2CO3 with the direction of the reaction entirely dictated by pH (see Henderson Hasselbalch equation).
Methods:
L: 176-177: Is it possible that no chamber was produced at all in the presence of AZM? This would lead to strong misinterpretations of the results from the LA-ICP-MS analyses. It would be very helpful if the authors could provide a prove that the regions alalyzed by LA-ICP-MS were generated under the presence of AZM (e.g. calcein labeling).
Discussion:
L275: „…conversion between bicarbonate and carbonate by Carbonic anhydrase…“See earlier comment. This is not correct. CA catalizes the the conversion from CO2 to H2CO3 dictated by pH.
283: „pumped out of the site of calcification and into the surrounding seawater…“ I find this sentence very confusing. If protons are pumped out from the site of calcification (i.e. intracellular vesices or the extracellular calcification space) they will first be inside the cell. Only in a second step protons may be transported out of the cell leading to the observed extracellular acidification. Please be more specific about the proposed compartments (i.e. SOC, cytosol and seawater) and proton gradients.
286: „…towards CO32- inside the the foraminifera…“ What is inside the foraminifera cytosol or SOC? What is the cytosolic pH in this foraminiferan species? This information is essential for the proposed CO2 fluxes.
L302: „…this suggests that foraminifera do not rely on [CO32-] itself…“ But they could rely on [HCO3-] that will increase under hypercapnic conditions. Along these lines an extracellular carbonic anhydrase has been characterized in sea urchin calcifying cells that promotes a carbon concentration mechanism (Matt et al. 2022 PNAS 4:119). This reference might be very useful for the present manuscript as it also explains enhanced calcification under elevated PCO2 conditions.
L304: „…with the active regulation of pH by foraminifera.“ Where is pH actively regulated? In the cytoplasm and / or in the clcification space?
L339-341: „Toyofuku et al. (2017) hypothesized that in rotalid foraminifera, the outward proton pumping is proportional to the inward Ca2+ flux…“ If the exchange of H+ against Ca2+ is proportional the ability to concentrate Ca2+ will be insufficient to fuel high calcification rates. This is due to the extremely low [H+] in the sea water/boundary layer compared to [Ca2+]. While [Ca2+] in seawater is in the rage of 10 mM The concentration of [H+] at a pH of 8 is only 0.00001mM and at a pH of 6 (given strong boundary layer acidification) 0.001mM. I do not see how proton Ca2+ exchange in the nM to µM range should significantly contribute to Ca2+ uptake for mineralization.
L341-343: „The inward diffusion of CO, due to the steep inward-outward pH gradient……provides the necessary protons to maintain the exchange of two protons (out) for one proton (in).“ This sentence requires clarification. First, what is „in“ and what is „out“. Does „in“ refer to the cytoplasm or the calcification space?
According to the authors a boundary layer acidification is generated by proton export from the cytoplasm. To generate a higher pCO2 in the boundary layer than in the cytoplasm, the pH in the boundary layer must be lower than the cytoplasmic pH. To validate their argumentation the authors must provide the exact pH conditions in the boundary layer, the cytoplasm and the calcifcation space.
344-349: In this part of the discussion the authors can refer to a recent study (Dubicka et al. 2023 Heliyon 9:e18331), that used life imaging techniques to demonstrate the uptake and modulation of Ca2+ and Mg2+ in foraminifera vesicles during calcification. In their model Mg2+ is removed from Ca2+ rich vesicles shortly before deposition into the skeleton.

Cite this review as

---

## Round 0.2 · Minor Revisions

· Academic Editor

Minor Revisions

All three of the original reviewers have now provided their feedback on the revised manuscript.They are recommending publication, yet all offer a few more suggestions. Following their assessment, I would like to ask you to address their comments, and either rebut their feedback or revise the manuscript accordingly as part of a minor revision. I personally agree that plotting a Rayleigh model (reviewer 2) could benefit the paper, but ultimately consider this up to the authors, and look forward to seeing this manuscript in print.

Reviewer 1 ·

Basic reporting

This manuscript by De Goeyse et al. reports growth rate and shell chemistry of benthic foraminifera Amphistegina lessonii cultured under carefully controlled environments. They investigate the influences of (1) enzyme carbonic anhydrase (CA) and (2) pCO2 on the shell formation process by comparing specimens incubated with/without a CA inhibitor and under various high pCO2. The results show that the inhibitor reduces calcification greatly, while pCO2 has a small but positive effect on calcification. Together with the shell chemistry analysis on Mg, Na, and Sr, the authors hypothesize a coupling between inorganic carbon uptake and Ca pumping at the site of calcification, which is an exciting demonstration of biological controls on foraminifera biomineralisation. The questions raised in my previous review are attentively addressed by the authors, and the manuscript is logically structured to present their evident results which are valuable to the biomineralisation community. I would be pleased to recommend this manuscript for publication after three minor suggestions as listed below.

L221-223: The t-tests here demonstrate the consistency between replicates of one condition. Here it is written “For TA, two pairs of replicates had slightly different average values…; for the DIC analysis, one pair within a treatment had significantly different average values’’. It would be clearer to add which pairs are the different ones as mentioned in the answers to reviewers document. Nevertheless, I do agree and think it would worth mentioning in this paragraph that all TA and DIC are similar within replicates, considering their small variations under each condition.

Fig 4: The colour is less ideal. It is not easy to see the green/brown colours in the plot. Is there only one green-brown colour, or are there two green and brown colours to represent the control groups? If it is the latter, what is the difference between green and brown data points?

Table S1: I would suggest adding calculated [CO3]2- in the table, as in L333-336 it argues that foraminifera do not rely on [CO3]2- for calcification (which I totally agree this is what the data suggests). It might be more intuitive to show a decrease in [CO3]2- with increasing pCO2 in the culture settings.

I would be happy to review the manuscript again, although I would be surprised if this were deemed necessary given the nature of my comments.

Experimental design

no comment

Validity of the findings

no comment

Additional comments

no comment

Cite this review as

Reviewer 2 ·

Basic reporting

Please see my review in the additional comments section

Experimental design

Please see my review in the additional comments section

Validity of the findings

Please see my review in the additional comments section

Additional comments

This is my second review of the manuscript “The effect of carbonic anhydrase on foraminiferal Mg/Ca” by De Goeyse et al., which presents growth rate and major/trace element data for a species of foraminifera grown under four different atmospheric CO2 cultured both with and without a carbonic anhydrase inhibitor (acetazolamide).

The authors have addressed the majority of the review comments and the manuscript is substantially improved as a result, in particular, as the impact of AZ on calcification versus ion transport is now more clearly separated. As such, while I have some further minor comments and would still suggest rephrasing in a few places to make this distinction even clearer, I again congratulate the authors on a really nice study and look forward to seeing it published.

My only slightly more substantive comment is that there are some very simple additional analyses that could be performed to test the key hypotheses in the later part of the discussion. In the response document the authors argue that including a Rayleigh distillation model is ‘beyond the scope of this paper’ but I don’t understand why given that this is really not a lot of work (it is one equation). Plotting a Rayleigh model on Fig. 4 as well as the slope of a mineralogical control would be really useful as it would enable the competing mechanisms discussed on line 420-440 to be distinguished in a less qualitative manner. I suspect you’ll find that the mineralogical slope is almost the same as the regression line through the data, whereas the Rayleigh model isn’t.

Specific comments:
• Line 77. Ammonia might be characterised by an external pH of 6.5 (during calcification) but this isn’t the case for Amphistegina (Glas et al., 2012).
• Line 79. Clarify what you mean by ‘occasional’.
• Line 87. Fix ‘ref’.
• Line 90. It might be the rate limiting step in terms of carbon concentration but I’m unaware that the study cited here constrains its role in limiting calcification in general.
• Figure 1. On reflection, I don’t think I understand why this is a figure rather than a table. It doesn’t show anything except a repeating clipart image of a culture jar.
• Line 176. Please add a note to the main text somewhere stating approximately how many chambers were added. The cross sectional area growth rate data is great, but this is essential information in terms of understanding whether the TE data are derived from chambers grown in culture.
• Line 184. Fix ‘with a in an’.
• Line 185. Space missing.
• Line 190. No possessive apostrophe needed.
• Lines 191-193. I didn’t understand how both the NIST glasses and NFHS2 are both used for standardization. Why is an additional drift correction needed? Weren’t the NIST glasses analyzed at regular intervals? Please explain properly.
• Lines 194-195. Please also state accuracy and how it was determined.
• Line 240. Superscript 2+.
• Lines 310-311. What is the basis for this statement? Please explain or delete.
• Line 352. Is it OK to cite an in prep. manuscript?
• Lines 355-357. This statement could benefit from being more fully explained. Also note the typo (therefore).
• Lines 367-369. You do now give an alternate hypothesis in the next paragraph, but I’d nonetheless caution against phrasing the ion transport section in these terms. You cannot distinguish between a difference in ion transport and a difference in the ratio of primary to secondary calcite. At the very least, I would caveat ‘indicates..’ with ‘assuming an equivalent proportion of primary and secondary calcite in the shell walls’.
• Lines 388-389. Again, the pH gradient is nowhere near as high as this in this species.
• Lines 402-404. You should also mention Mg2+ transport here. And note that Ca2+ transport will not substantially ‘lower the concentration of Mg2+’, although it would lower the Mg/Ca ratio.
• Lines 420-422. I would test it! Add the Rayleigh trend to the figure. As it stands, this very short discussion paragraph is unconnected to the one that follows it.
• Lines 429-431. This is factually incorrect, please remove or rephrase. In the case of Mg transport you would expect a mineralogical impact on Na/Ca as you later discuss, and which has a slope almost exactly as that in Fig. 4. I would also take issue with the use of the word ‘seems’ in the next sentence, based on the data presented here.
• Line 447. If you mean a plot of [CO32-] versus Mg/Ca, then the slope is negative, not positive.
• Line 454. Ca or Mg pumping rates. Or in fact, a different proportion of primary versus secondary calcite.

Cite this review as

Reviewer 3 ·

Basic reporting

Partly.

Experimental design

Yes.

Validity of the findings

The authors have addresses some of my previous concerns, but failed to provide a clear answer to the proton gradients across the different compartments from the site of calcification to the cytoplasm and finally to the seawater. However, this information is essential for the proper interpretation of their calcification model and the contribution of the carbonic anhydrase. I carefully studied the papers of Erez (2003) and Bentov and Erez (2006) that describe the morphology of the calcification compartment and the surrounding cell. Based on these studies alkaline conditions at the calcification front must be generated by a transport of protons from the SOC through the cytoplasm to the extracellular environment. The authors refer to some of their previous publications (i.e. De Nooijer et al. (2009), Glas et al. (2013), Toyofuku et al. (2017) with regard to the proton gradients across the different compartments, but I could not find any information on the cytosolic pH in these studies. Using the membrane impermeable dye HPTS the authors could demonstrate pH conditions in endocytotic vesicles but not the cytoplasm. However, this is essential to support and explain the proposed model of inward directed CO2 gradients. I think the authors should add a schematic drawing to this study, that clearly indicates the different compartments ( SOC, Cytoplasm and extracellular space ), including their pH conditions, proposed CO2 partial pressures, as well as the sites where carbonic anhydrase is proposed to catalize the pH dependent hydration of CO2. This information is critical for the reader to be able to precisely follow the argumentation of CO2 gradients and involvement of CA presented in the present work as well as previous studies on pH regulation in foraminifera.

Cite this review as

---

## Round 0.3 · Minor Revisions

· Academic Editor

Minor Revisions

I have now received the review of your revised manuscript. In the reviewer's assessment the revision addressed the review feedback and recommends the manuscript to be accepted. But the reviewer also has some final additional comments, which I would like to ask you to consider, in particular the one related to Figure 4, where they suggest clarifying the distinction between observations and conjectured processes. I believe that this will be particularly helpful for a broad and diverse readership that most benefits from this conceptual figure. Hence I hope you can add some clarifications along those lines in figure 4 and/or its caption.

Reviewer 2 ·

Basic reporting

Please see below

Experimental design

Please see below

Validity of the findings

Please see below

Additional comments

The authors have, in my view, addressed all of the outstanding review comments and the manuscript could be published in its present form. While we could debate the details further, no single contribution can be expected to address all outstanding issues and I think the authors have made a clear attempt to present a balanced discussion of the results in this iteration. I have just three further minor comments on re-reading the manuscript, one of which is more important and relates to the new figure, while the others could be addressed at the discretion of the authors and editor. Otherwise, I enjoyed reading the manuscript once more and congratulate the authors again on a nice piece of work.

- line 194-200. If you calibrate NFHS-2-NP to NIST SRM610 then the drift correction is ultimately anchored to the NIST glass, and I still don't understand the need for both to feature in the data processing. I likely misunderstand. You could add the equations to the supplement to make sure that others don't as well, or state where these can be found.

- lines 450-452. I still disagree about pH gradients in this species (it's probably elevated due to photosynthesis, not reduced), but even leaving this aside there is no evidence for pH = 6 in the boundary layer of any foraminifer.

- figure 4. As per earlier comments, this is THE species for which excellent observations of vacuolisation exist. They must feature on any schematic figure. It would also be worth pointing out in the figure caption that any pH gradient depends on the balance between proton pumping, respiration, and photosynthesis, and these details remain unknown for this species (and indeed any symbiont-bearing species), particularly in terms of when calcification versus P/R take place (whether this happens during the day or night will be key, for example). The existence of a pH gradient during shell formation remains entirely conjectural for this species and this should be clear - a 'model of calcification in foraminifera' missing key processes is not necessarily a model that is usefully applicable to the species under consideration here.

Cite this review as

---

## Round 0.4 · accepted · Accept

· Academic Editor

Accept

Thank you for the revisions in the text and Figure 4. Your responses address the final comments and I look forward to seeing your manuscript in print.